# A Preclinical Investigation on the Role of IgG Antibodies against Coagulant Components in Multiple Sclerosis

**DOI:** 10.3390/biomedicines11030906

**Published:** 2023-03-15

**Authors:** Maria S. Hadjiagapiou, George Krashias, Elie Deeba, Christina Christodoulou, Marios Pantzaris, Anastasia Lambrianides

**Affiliations:** 1Department of Neuroimmunology, The Cyprus Institute of Neurology and Genetics, Nicosia 2410, Cyprus; 2Department of Molecular Virology, The Cyprus Institute of Neurology and Genetics, Nicosia 2410, Cyprus

**Keywords:** multiple sclerosis, inflammation, coagulation, antibodies, astrocytes, signalling pathways, Luminex xMAP technology

## Abstract

The coagulation-inflammation interplay has recently been identified as a critical risk factor in the early onset of multiple sclerosis (MS), and antibodies against coagulation components have been recognized as contributing factors to thrombotic and inflammatory signaling pathways in diseases with overlapping symptoms to MS, paving the way for further research into their effects on MS pathology. The current study aimed to enlighten the role of IgG antibodies against coagulation components by performing a preclinical study, analyzing the astrocytic activation by purified IgG antibodies derived from 15 MS patients, and assessing their possible pro-inflammatory effects using a bead-based multiplexed immunoassay system. The results were compared with those obtained following astrocyte treatment with samples from 14 age- and gender-matched healthy donors, negative for IgG antibody presence. Serum samples collected from 167 MS patients and 40 age- and gender-matched controls were also analyzed for pro- and anti-inflammatory factors. According to our results, astrocytic activation in response to IgG treatment caused an upregulation of various pro-inflammatory factors, including cytokines, chemokines, and interleukins. Conversely, in serum samples from patients and controls, the pro-inflammatory factors did not differ significantly; medication may lower the levels in patients. Our findings suggest that antibodies may function as effectors in neuroinflammation and serve as targets for new treatments that eventually benefit novel therapeutic approaches.

## 1. Introduction

Multiple sclerosis (MS) is a heterogeneous neurological disease that occurs primarily between 20 and 50 years of age and causes severe cognitive and physical impairment [1,2]. It is characterized as a chronic, autoimmune inflammatory disease, affecting the white and gray matter of the central nervous system (CNS) and producing multifocal plaques as a result of local and infiltrated reactive immune cell-mediated inflammation. The myelin sheath surrounding nerve axons is destroyed, and impulse transmission through the axons is disrupted, leading to axon degeneration and loss [1,2].

Although the pathogenesis of MS remains unknown, insufficient regulation of the coagulation-inflammation circuit has recently been proposed as a potential factor in triggering inflammatory responses and being a key player in chronic neuroinflammation [3,4]. Interestingly, activated clotting factors FXIIa and FXIIIa have been detected in MS cases, especially in RRMS and SPMS patients [5], with the latter implicated in the blood-brain barrier (BBB) increased permeability and upregulation of adhesion molecules, leading to the adhesion of leukocytes to endothelial cells. Moreover, FXIIa and FXIIIa have been involved in modulating the structure of monocytes, remodeling their cytoskeleton, and further enhancing the phagocytosis process. Simultaneously, elevated FXIIa expression levels have been associated with a high risk of relapse [6].

Importantly, the deposition of thrombin and fibrin(ogen) in the CNS of MS animal models precedes the onset of neurological manifestations, while they are detected in areas where scar lesions will appear [7]. In the brain, parenchyma fibrin(ogen) has been shown to modulate and attract microglia, macrophages, monocytes, and T-cells in pre-demyelinated regions, causing inflammatory responses following the overexpression of chemokines and cytokines such as CXCL10, CCL2, and IL-12 by the immune cells [8]. Notably, fibrin(ogen) acts through its binding to integrin receptors expressed on microglia and T cell surfaces, i.e., beta-1 (alpha5-beta-1), beta-2 (CD11b/CD18 and CD11c/CD18), and beta-3 (alpha V-beta-3) [9]. The generation of fibrin monomers by the fibrinogen precursor results in the unmasking of the γ377-395 epitope in the fibrin molecule, which is essential for the recognition and association with the CD11b/CD18 microglia integrin receptor [10]. In turn, the fibrin-integrin receptor complex can induce the production of reactive oxygen species (ROS) and the overexpression of pro-inflammatory molecules such as TNF-a, NF-kB, and IL-1a by microglia and macrophages, maintaining the inflammatory milieu within the CNS [11]. Fibrin(ogen) deposits are detected in acute CNS plaques and co-localize with active astrocytes, microglia, invasive macrophages, and T cells in reactive MS lesions [3]. Their presence is frequently demonstrated in the deepest cortical layers, V and VI, contributing to neurodegeneration and axonal loss [12]. In addition, thrombin can stimulate the activation of complement components, and attracts the microglia’s mobility in areas where the myelin sheath will eventually be destroyed, enhancing inflammatory responses [13].

Since serine proteases of the coagulation system have attracted attention not only as coagulant mediators but also as potential effectors of inflammation, it is of primary importance to investigate the presence of antibodies directed against coagulant serine proteases that may be responsible for the initiation and progression of autoimmune and inflammatory diseases such as MS.

Few studies have been published regarding the presence of antibodies associated with the coagulation cascade and thrombosis in autoimmune diseases [14]. So far, antibodies against thrombin, PC, plasmin, and FXa have been detected in systemic lupus erythematosus (SLE) and antiphospholipid syndrome (APS), allowing researchers to characterize their role in the underlying pathways that lead to disease onset [15]. In APS studies, for instance, reactivity to activated protein C (APC) has been shown to inhibit the protein C/APC anticoagulant mechanism and interfere with the suppression of FVa and FVIIIa functions. Thus, such antibodies potentiate rather than inhibit thrombotic and procoagulant effects [15,16].

In our previous work, we achieved to show that an increased proportion of patients diagnosed with MS (43%) were positive for the presence of IgG antibodies against seven coagulation serine proteases, i.e., factor VIIa (FVIIa), thrombin, prothrombin, FXa, FXII, protein C, and plasmin, also demonstrating that such molecules can be correlated with advance disability status and progression of the disease [17].

Considering that patients with MS have an increased risk of deep vein thrombosis, pulmonary embolism, ischemic stroke, and cardiovascular disorders in the early onset of the disease [18], it is of great importance to understand the role of IgG antibodies against coagulation factors in the context of MS pathophysiology. Therefore, the following objectives have been established for the current research study:Identification of the role of IgG antibodies against coagulation components in MS. Following a series of in vitro experiments with purified IgG antibodies, the role of such molecules will be characterized regarding the changes in the expression levels of inflammatory mediators.Analysis of the expression levels of inflammatory and neuroinflammatory mediators in MS patients and comparison with controls.

## 2. Materials and Methods

### 2.1. Study Participants

Fifteen MS patients and 14 age- and gender-matched healthy participants (*p* < 0.95 and *p* = 1.0, respectively) were recruited for the preclinical investigation of IgG antibodies against coagulation components. As previously reported [17], all patients tested positive for antibodies against coagulant serine proteases, whereas controls tested negative. Serum samples were also collected from 167 MS patients and 40 age- and gender-matched HCs for the purpose of screening and quantifying pro- and anti-inflammatory factors of interest in a large cohort of patients and comparing the results with those obtained from HC samples, as well as from our in vitro experiments. The main diagram of the methodology that was followed for the current research study is presented in Figure 1.

All patients were followed up in the Department of Neuroimmunology at the Cyprus Institute of Neurology and Genetics between September 2017 and January 2019, satisfying all McDonald’s revised criteria for inclusion [19]. They also strictly adhered to the following guidelines: age above 18 years old; patients with clearly identified classification [Relapsing-Remitting MS (RRMS), Secondary Progressive (SPMS), Primary Progressive MS (PPMS)]; sufficient demographic and clinical data (age of onset, expanded-disability status scale (EDSS), medication, other autoimmune and non-autoimmune disorders). Excluded criteria were pregnancy, drug, and alcohol abuse. The participants signed a written consent form approved by the Cyprus National Bioethics Committee (ΕΕΒΚ/ΕΠ/2016/51).

### 2.2. Purification of IgG Antibodies against Coagulant Components from Serum Samples Using Affinity Chromatography

Serum samples derived from MS patients with seropositivity status against factor (F)VIIa, thrombin, prothrombin, FXa, FXII, plasmin, and protein C were subjected to the purification procedure by affinity chromatography, using the Nab Protein G Spin Columns (ThermoFisher Scientific, Waltham, MA, USA). Samples from HCs were also subjected to the same procedure to be used as negative controls in the in vitro studies.

For the purification process, gravity flow was employed. Unless stated otherwise, all buffers were prepared according to the manufacturer’s instructions. Serum samples were diluted 1:1 with binding buffer (0.1 M phosphate, 0.15 M sodium chloride; pH 7.2). Protein G columns were equilibrated with the binding buffer, and the serum samples passed through the columns. Purified antibodies were eluted using an acidic elution buffer (0.1 M glycine, pH 2–3) and collected in Eppendorf tubes containing 100 mL of neutralization buffer (1 M Tris at pH 8–9). Subsequently, the columns were regenerated with elution buffer and stored at 4 °C in PBS containing 0.02% sodium azide. Using an Amicon Ultra-15 centrifugal filter unit, the IgG fractions from each sample were pooled and collected in 1 mL of PBS.

All purified samples were then subjected to an endotoxin removal procedure to remove any endotoxin trace, known to stimulate the inflammatory response and activate the innate immune system. The Pierce High Capacity Endotoxin Removal Resin (ThermoFisher Scientific, Waltham, MA, USA) was applied, and the buffers were prepared as directed. Following the manufacturer’s recommendations, the resin was regenerated by adding five volumes of 0.2 N NaOH in 95% ethanol for 1–2 h at RT. Columns were washed with 2 M NaCl and ultrapure water five times and then equilibrated with sterile endotoxin-free PBS and sodium phosphate buffer at pH 6–8. In the following step, purified IgG samples were added at a flow rate of 0.25 mL per minute, and the flow-through was collected in an endotoxin-free tube. Immunoglobulins were eluted with sterile, endotoxin-free PBS. Then, the presence/absence of endotoxin was determined by the *Limulus* Amoebocyte Lysate assay (Sigma, Dorset, UK), in which all purified samples were tested negative for the endotoxin presence (<0.06 endotoxin units/mL considered to be endotoxin-free).

### 2.3. Activation of Astrocytic U87 Cell Line

The human U87 astrocytic cell line (Cell Line Service, Eppelheim, Germany) was used as an experimental model to examine whether IgG antibodies against coagulant components are involved in MS pathology. The cells were cultured at a density of 2 × 10^4^ cells/cm^2^ in Minimum Essential Medium (MEM) supplemented with 10% fetal bovine serum, 5 mM penicillin-streptomycin, 2 mM L-glutamine, 0.1 mM non-essential amino acid, 1 mM sodium pyruvate, and the cultured flasks were pre-treated with Nunclon^TM^ Delta to ensure consistent growth of the cells.

The stimulation process for the U87 astrocytic cell line was performed in six-well plates, where cells were allowed to be seeded at a density of 5 × 10^5^/mL on the pre-treated bottom of the wells. In each vessel, 1 mL of fresh medium was added, along with the appropriate concentration of stimuli. An amount of 100μg/mL of endotoxin-free purified IgG derived from the patients and HC-purified samples were used to treat U87 cells. Five units/mL of thrombin (Sigma-Aldrich, St. Louis, MI, USA) and 100 ng/mL of TNF-a (ThermoFisher Scientific, USA) were used as positive controls for the experiment procedure, while untreated U87 cells were used as the reference sample.

### 2.4. Lysis of Cells

Radioimmunoprecipitation assay (RIPA) lysis buffer (1X: 50 mL 1 M Tris; pH 7.4; 4 M NaCl; 0.5 M EDTA; NP-40; 10% SDS; d.d. H_2_O) was added to each well containing untreated or treated U87 cells with IgG stimuli to enable cell lysis. Inhibitors of protein phosphatases (NaF and Na_3_VO_4_), as well as protease inhibitors (Roche, Mannheim, Germany), were also added to the RIPA prior to the use. After 30 min of RIPA incubation on ice, tubes containing astrocytic lysates were frozen-thawed several times in liquid N-2 to further support the lysis. Centrifugation at 14,000 rpm was performed for 15 min, and the supernatant containing proteins was collected in new tubes.

### 2.5. Analysis of Pro- and Anti-Inflammatory Factors Performing Multiplexed Immunoassay System

The human inflammatory 20-Plex Panel was analyzed by multi-analyte profiling (xMAP) Luminex technology (TheromoFisher, USA) in the IgG–stimulated and unstimulated U87 cells, as well as in cells stimulated with HC samples. In addition, serum samples of MS patients and HCs were also screened and analyzed for the same panel. Namely, GM-CSF, IFN alpha, IFN gamma, IL-1 alpha, IL-1 beta, IL-4, IL-6, IL-8, IL-10, IL-12p70, IL-13, IL-17A (CTLA-8), TNF alpha, CXCL10, CCL2, CCL3, CCL4, ICAM-1, E-selectin, P-Selectin were simultaneously detected and quantified.

As instructed by the manufacturer, the procedure began with the reconstitution of antigen standards with a standard buffer. In accordance with the procedure, magnetic beads were added to each well of the 96-well plate, and the plate was left on a magnetic plate washer to allow the beads to cover the bottom of the wells, and the liquid was then removed. As per the design, a 1:1 standard buffer along with controls, samples tested, and the negative control (buffer) were distributed into each well, following incubation at RT. Subsequently, an antibody detection mix was incubated in each well at RT. For signal development, streptavidin conjugated with phycoerythrin (PE) was added, and the plate was incubated under the same conditions as previously (RT). Then, the beads were re-suspended using the Reading buffer, incubated at 800 rpm, and RT for five minutes, and the results were finally analyzed on the MagPixTM system.

### 2.6. Statistical Analysis

The analysis of the results was carried out using GraphPad Prism V.5.00 for Windows (La Jolla, CA, USA). The D’Agostino-Pearson test was used to test for normality [20]. The non-parametric Mann-Whitney U-test was applied to analyze age-matching data and to compare differences in expression levels of analytes between the two studied groups [21]. Kruskal–Wallis one-way analysis of variance (ANOVA) followed by Dunn’s post hoc test for multiple comparisons was used to assess the pro-inflammatory mediator’s presence in more than two groups of participants [22]. Fisher’s exact test was performed to assess the matching of gender among the study groups [23]. The accuracy of the test was evaluated using the area under a Receiver Operating Characteristic (ROC) curve (AUC) [24]. A correlation matrix was performed using the Spearman correlation coefficient to analyze the association between antibody activity levels [25]. The values range from −1, which corresponds to the perfect negative correlation, to +1, which is the perfect positive correlation.

## 3. Results

### 3.1. Characterization of the Role of IgG Antibodies against Coagulation Components

#### 3.1.1. Purified IgG Antibody Fractions

A preliminary investigation was conducted to characterize the role of IgG antibodies against seven coagulant components, i.e., FVIIa, thrombin, prothrombin, FXa, FXII, plasmin, and protein C, and to determine whether these molecules could potentially serve as pro-inflammatory biomarkers in MS pathology. Purified IgG antibodies were derived from serum samples of 15 MS patients who tested positive for at least one IgG antibody, while samples from 14 healthy controls who tested negative for all antibodies studied were used as controls.

Following purification of the aforementioned IgG antibodies, the IgG activity was confirmed in all purified fractions by enzyme-linked immunosorbent assay (ELISA) (Table 1), using a concentration of 100 μg/mL from each purified fraction tested. The initial ELISA protocols described in the methodology of our previous work were performed [17].

#### 3.1.2. Activation of Intracellular Pathways upon Astrocyte Stimulation with IgG Antibodies

Quantification analysis of the inflammatory 20-plex panel included a variety of anti-inflammatory and pro-inflammatory molecules, namely GM-CSF, IFN alpha, IFN gamma, IL-1 alpha, IL-1 beta, IL-4, IL-6, IL-8, IL-10, IL-12p70, IL-13, IL-17A (CTLA-8), TNF alpha, CXCL10, CCL2, CCL3, CCL4, ICAM-1, E-selectin, P-Selectin was performed upon in vitro activation of astrocytes with purified IgG fractions. As shown in Table 2, astrocytes stimulated by purified IgG samples from MS patients had a significantly higher mean level of concentration for most of the molecules studied than HCs. In particular, significantly higher mean levels of the E-selectin adhesion molecule were revealed after cell activation by MS-IgG fractions [mean: 1089 (standard error of the mean; SEM: 28.12)] than HCs [mean: 702.3 (SEM: 15.90); *p* < 0.0001]. Similarly, the ICAM adhesion molecule was overexpressed in MS-stimulated astrocytes [mean: 10,790 (SEM: 586.4)] compared to HCs [mean: 9087 (SEM: 616.3); *p* = 0.0137]. Concentration levels of Il-1a and IL-1b were also elevated upon stimulation with patient-purified antibody samples [IL-1a mean: 13.30 (SEM: 0.5678); IL-1b mean: 3120 (SEM: 197.9), respectively] than controls [IL-1a mean: 9.574 (SEM: 0.4023) *p* < 0.0001; IL-1b mean: 2706 (SEM: 115.0), *p* = 0.0094, respectively], and this was also true when analyzing the levels of the chemokines studied. Namely, CXCL10, CCL2, CCL3, and CCL4 were secreted at higher levels upon stimulation with MS-IgG fractions [CXCL10 mean: 2.807 (SEM: 0.1334); CCL2 mean: 17.08 (SEM: 0.4888); CCL3 mean: 380.8 (SEM: 14.69); CCL4 mean: 50.30 (SEM: 1.294)] compared to HC fractions [CXCL10 mean: 2.001 (SEM: 0.2249) *p* = 0.0042; CCL2 mean: 13.13 (SEM: 0.5012), *p* < 0.0001; CCL3 mean: 296.8 (SEM: 15.87), *p* = 0.0013; CCL4 mean: 39.19 (SEM: 1.708), *p* < 0.0001]. Moreover, the pro-inflammatory factor TNF-a was also upregulated when astrocytes were incubated with purified IgG antibodies from MS patients [mean: 22.37 (SEM: 0.6531) vs HCs mean: 15.77 (SEM: 0.3263); *p* < 0.0001].

When analyzing the accuracy of the test, the area under the curve (AUC) value was found to be significantly increased, which corresponds to effective discrimination of the pro-inflammatory mediators’ upregulation due to IgG antibodies compared to negative controls. The AUC value provides substantial evidence for the perfect accuracy of the test. Figure 2 shows the receiver operating characteristic curve (ROC) with the AUC for the pro-inflammatory factors that were significantly elevated in MS patients compared to HCs.

#### 3.1.3. Correlation between the Concentration Levels of Pro-Inflammatory Mediators

Correlation analysis between the concentration levels of the analytes studied by Spearman correlation coefficient revealed monotonically negative to monotonically positive ranks. Specifically, a moderate negative correlation between ICAM-1 concentration and GM-CSF (r_s_= −0.52, *p* < 0.05) (Figure 3A) was reported, in addition to a strong negative correlation between ICAM-1 and IL-8 concentration levels (r_s_= −0.66, *p* < 0.01) (Figure 3B).

Ranks of positive moderate and positive, strong correlations of pro-inflammatory mediators were also revealed, indicating that such molecules can interplay, enhancing the inflammatory milieu. Emphasis is given to the strong positive correlations. In particular, the levels of the E-selectin and P-selectin adhesion molecules were correlated, showing a high rank (r_s_ = 0.85, *p* < 0.0001) (Figure 4A), and this was also true when analyzing the levels between GM-CSF and IL-10 (r_s_ = 0.85, *p* < 0.0001) (Figure 4B). A significantly strong correlation was also revealed between the IL-1a and IL-1b concentration levels (r_s_ = 0.83, *p* < 0.001) (Figure 4C), IL-1a and CCL3 (r_s_ = 0.82, *p* < 0.001) (Figure 4D), and between IL-1a and CCL4 (r_s_ = 0.82, *p* < 0.001) (Figure 4E). Likewise, the levels of produced pro-inflammatory IL-1b by activated U87 cells were strongly correlated with those of chemokines CCL3 (r_s_ = 0.86, *p* < 0.001) (Figure 4F) and CCL4 (r_s_ = 0.85, *p <* 0.001) (Figure 4G) and similarly, a high rank of correlation revealed for the CCL3 and CCL4 levels (r_s_ = 0.97, *p* < 0.001) (Figure 4H). These correlations are also illustrated in a heat map (Figure 5), in addition to the correlation between all the analytes concentration levels, giving the different ranks of the correlation coefficient.

### 3.2. Evaluation of Inflammatory Factors in Serum Samples of MS and HC Participants

#### 3.2.1. Data of Study Participants

Analysis of pro- and anti-inflammatory factors in serum samples of 167 MS patients and 40 age- and gender-matched HCs was also conducted, quantifying the concentration levels of the aforementioned molecules included in the human inflammation 20-Plex ProcartaPlex panel (ThermoFisher, USA). Table 3 provides data on the clinical and non-clinical characteristics of the participants.

#### 3.2.2. Detection and Quantification of Pro- and Anti-Inflammatory Mediators in Patients and HCs

According to the analysis, almost all the molecules studied showed no significant differences between the patients and controls. Nevertheless, four pro-inflammatory factors differed significantly between the two groups showing increased levels in HCs than in patients. Specifically, we detected higher levels of GM-CSF in HCs [mean: 8.74 (SEM: 2.08)] than in MS [mean: 7.33 (SEM: 2.40); *p* = 0.0102], and this was also true when analyzing the levels of IL-1a in the control group [mean: 6.29 (SEM: 1.09)] compared to the patient group [mean: 4.85 (SEM: 0.65); *p* = 0.0012]. Likewise, the concentration of IL-8 was increased in the HCs [mean: 84.97 (SEM: 19.21); mean of patients: 55.59 (SEM: 12.34); *p* = 0.003], as well as the concentration of CCL4 [mean: 193.4 (SEM: 19.87)] compared to MS patients [mean: 150.9 (SEM: 10.48); *p* = 0.0043] (Table 4).

An in-depth analysis based on the disease courses, and especially relapsing-remitting (RRMS) and secondary progressive types of MS (SPMS), revealed that RRMS patients instead of SPMS showed significantly lower concentration levels of IL-1a, IL-8, and CCL4 compared to HCs (Figure 6).

In particular, RRMS patients had significantly decreased levels of concentration for IL-1a [mean: 4.27 (SEM: 0.67)]; IL-8 [mean: 47.85 (SEM:13.01)]; and CCL4 [mean: 139.4 (SEM:10.18)] than the controls [IL-1a mean: 6.29 (SEM: 1.06); *p* < 0.05]; [IL-8 mean: 84.97 (SEM: 19.21); *p* < 0.01]; [CCL4 mean: 193.4 (SEM: 19.85); *p* < 0.01] (Table 5); however, this difference was not observed when comparing SPMS patients and HCs or RRMS and SPMS patients. Patients diagnosed with CIS or PPMS were not included in the analysis due to the small sample size of the patients analyzed.

#### 3.2.3. Assessing the Pro-Inflammatory Profile of Patients Regarding Their Medication

Considering the prescribed treatment for MS patients, which included immunomodulatory and immunosuppressive drugs, as well as those patients without any medication, we examined the pro-inflammatory and anti-inflammatory factors related to the therapy that patients with MS received. According to our data, no difference was exhibited based on the type of medication for almost all the factors. Nevertheless, it was found that patients receiving immunomodulatory drugs had increased levels of CXCL10 concentration, which was significantly elevated when compared with patients receiving immunosuppressive drugs (*p* < 0.01), and this was also true when analyzing the concentration levels between immunomodulatory drugs and non-medication (*p* < 0.01) (Figure 7).

## 4. Discussion

In our preclinical study, the effects of IgG antibodies against procoagulant serine proteases in MS pathology provided substantial evidence that such antibodies can provoke signaling pathways contributing to MS underlying mechanisms. Interestingly, a wide range of pro-inflammatory mediators, interleukins, and chemokines was expressed at significantly elevated levels when astrocytes were activated by IgG fractions derived from MS patients positive for IgG antibodies against coagulant components, whilst astrocytes exposed to HC fractions did not exhibit similar levels. To our knowledge, this is the first study that examines the role of IgG against coagulation components in MS pathology and assesses signaling pathways that may be involved in the disease onset.

Recently, much attention has been paid to the characterization of the role of coagulant proteases involved in neurological diseases, considering their accumulation in pre-demyelinated regions of the CNS. Specifically, Davalos and his colleagues [7] showed that thrombin was capable of mediating pro-inflammatory effects in MS animal models, implicating itself in a variety of inflammatory responses. Moreover, thrombin can activate glial cells such as astrocytes and microglial cells to release pro-inflammatory mediators, such as TNF-a, IL-1b, IL-6, IL-8, GM-CSF, CCL2, CCL3, CCL5, and CXCL10 [26,27]. The non-coagulant effects of thrombin are mediated by thrombin-activated receptors, called protease-activated receptors (PARs), expressed on endothelial cells, monocytes, leukocytes, platelets, astrocytes, and neurons [28]. Namely, alteration in cellular structure, abnormally increased cell proliferation, production of nitric oxide, BBB leakage, expression of cyclooxygenase 2 (COX-2), and prostanoids are all consequences of PAR-1 activation by thrombin [29]. Therefore, suppression of thrombin-mediated PAR-1 activation may impede the abnormal signaling of PAR-1-activated pathways.

The coagulation-inflammation circuit is a dynamic system that regulates immune responses and maintains an inflammatory milieu under pathological conditions [30]. Notably, TNF-a, IL-6, IL-1, CCL3, CCL4, and CCL5 are all expressed on endothelial and fibroblast cells after PAR-1 signaling, as is ICAM-1, VCAM-1, E-, and P-selectin adhesion molecules, which are also expressed alongside MCP1 and PAI-1 and IB-factors [30].

So far, few studies have focused on the presence of antibodies against coagulant serine proteases in autoimmune diseases. In particular, antibodies directed against thrombin, PC, plasmin, and FXa have been evident in SLE and APS [15,31,32,33], which potentiate rather than inhibit thrombotic and procoagulant effects [15,16,34]. Earlier studies on APS indicated that such antibodies could prevent the inactivation of clotting factors, suggesting that they were able to show procoagulant effects. When the reactive center loop (RCL) of the antithrombin III anticoagulant molecule binds to the heparin-like glycosaminoglycan-rich domains of the clotting antigen, the protease cleaves the scissile P1-P1 bond of the RCL to form the P1-protease complex that results in the inactivation of the protease. Antibodies against coagulant components, on the other hand, can act as antagonists for antithrombin III binding capability to thrombin, preventing the formation of an antithrombin III-protease complex [35], inhibiting protease inactivation, and contributing to the thrombosis-inflammation interplay [36]. For instance, antibodies against FXa have been detected in patients diagnosed with SLE and APS, interfering with the antithrombin-mediated FXa inhibition and causing further thrombin generation [15].

Interestingly, activity against plasminogen/plasmin has also provided procoagulant effects that suppress the fibrinolytic pathway and further prolong the coagulation cascade [37]. Notably, antibodies against fibrinolytic enzymes appear to recognize and bind to epitopes of the protease catalytic region, preventing plasmin from binding to fibrin clots, and, as a result, fibrin clots are accumulated, leading to thrombosis [35,37].

Conversely, monoclonal antibodies against FXII suppress the activation of the intrinsic coagulant pathway mediated by FXII, resulting in anti-coagulant rather than coagulant effects [38,39]. In the Alzheimer’s disease (AD) mouse model, suppression of intrinsic coagulant pathways can reduce inflammatory responses and ameliorate the disease phenotype [38].

Our findings support that antibodies against coagulant serine proteases can also function as effectors of underlying mechanisms triggering inflammatory responses. All the purified IgG antibodies studied induced elevated concentration levels of pro-inflammatory factors to be released than samples from HCs, supporting the notion of coagulation-inflammation interplay as a result of insufficient regulation of the cross-talk between the coagulation cascade and inflammatory response. As far as we know, this is the first study to show high levels of secreted pro-inflammatory mediators following astrocyte activation by IgG fractions from MS patients who were found positive for procoagulant serine proteases. Since their presence prolongs rather than suppresses the coagulant cascade and thrombin generation, we can probably suggest that similar mechanisms are activated, especially the PAR-1 receptor, which contributes to the initiation of inflammatory responses. Furthermore, our evidence supports the hypothesis that MS-derived IgG antibodies against procoagulant serine proteases can act as pro-inflammatory stimuli, inducing the expression of E-selectin, GM-CSF, ICAM, IL-1a, IL-1b, IL-8, CXCL10, CCL2, CCL3, CCL4, and TNF-a in MS.

The pro-inflammatory and anti-inflammatory molecules studied in vitro were also analyzed in serum samples derived from patients with MS to quantify the concentration levels and make comparisons with those obtained from HC serum samples. Interestingly, there was no substantial difference between the studied groups for most of the factors of interest, whereas there were greater concentration levels for a few pro-inflammatory mediators in HCs than in patients. This observation leads to the speculation that medication contributes to the suppression of the immune response, hence the reduction in the production and release of key inflammatory molecules. Such speculation is also supported by earlier studies, which have shown higher IL-1 levels in HCs than in RRMS patients, suggesting that IFN-beta treatment might contribute to a decrease in IL-1 levels in MS [40]. Additionally, another study found higher IL-17a levels during relapse, along with an abundance of Th-17 cells; however, this was not true during the remission phase, and our patients were not in active relapse during the time of sampling [41].

The current study met some limitations that warrant discussion. An important limitation of our work was the small sample size of healthy controls compared to MS patients, which reduced the statistical power of our results. It is widely accepted that by having at least an equal sample size between the two groups of interest, the analysis could significantly increase the statistical power of the data obtained. Therefore, an increased sample size of patients and controls needs to be analyzed to produce more robust and valid data. Another limitation was the limited number of samples for the preclinical studies. Therefore, the cohort sample size should be expanded to confirm the existing results. Following our in vitro investigation of IgG antibodies against coagulation components, it would also be interesting to compare the effect of each antibody against coagulant serine proteases in neuroinflammation based on the evaluation of an increased number of purified IgG samples derived from MS patients. Since each patient’s antibody profile is unique, it is important to examine the effect of each antibody alone or in combination with other antibodies that target coagulation factors. It is also important to note that such experiments characterizing the role of antibodies directed against coagulation components should also be performed using primary human cells to validate our findings and assess them under conditions similar to those seen in vivo. Moreover, since the duration of the disease between the cases studied in our preclinical study varies, a future study is needed to investigate whether the duration of the disease could affect the results following a comparison between different periods of the disease based on IgG activity and expressed pro-inflammatory factors upon astrocytic activation with MS-derived IgG antibodies studied.

After assessing the role of IgG antibodies against coagulation components, we found convincing evidence that these antibodies may act as potential effectors of a wide range of signaling pathways and may enhance the coagulation-inflammation interplay in neuroinflammatory diseases such as MS. In view of these findings, further studies are required in order to provide new insights into the coagulation-inflammation circuit as well as to determine whether antibodies against clotting factors can serve as therapeutic targets.

## 5. Conclusions

The current study targeted the identification of the role of IgG antibodies against coagulant components in MS following astrocytic activation by purified IgG antibodies derived from MS patients. Overall, we have shown for the first time that IgG antibodies against clotting factors play a role in the development of an inflammatory milieu, and the presence of such antibodies can serve as a potential biomarker in neuroinflammatory diseases such as MS. Since the study of antibodies warrants further investigation, new findings will shed light on the underlying mechanisms that are activated and cause MS to develop, providing new insights into the disease’s prognosis and monitoring as well as the development of novel therapeutic strategies.

## Figures and Tables

**Figure 1 biomedicines-11-00906-f001:**
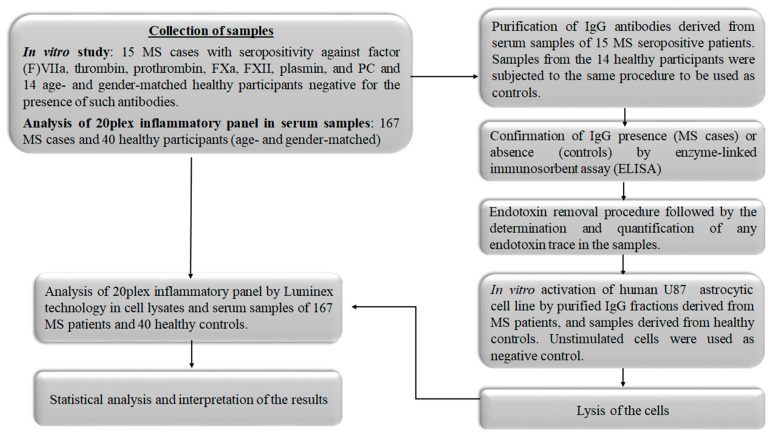
Diagram of research methodology.

**Figure 2 biomedicines-11-00906-f002:**
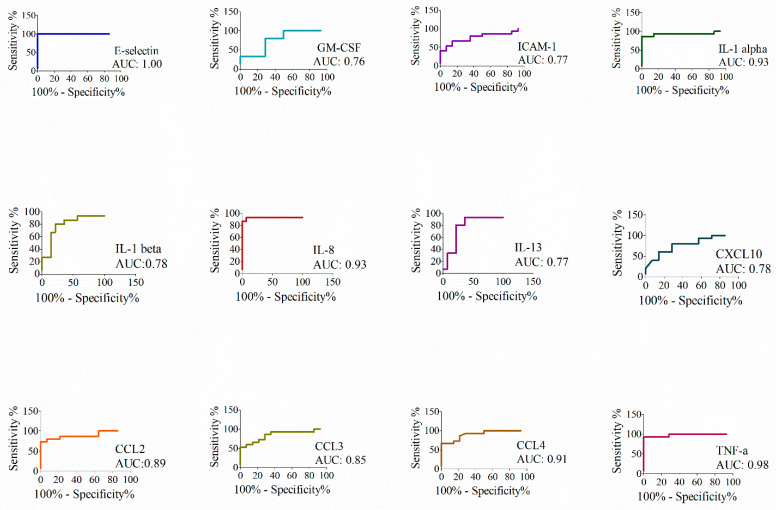
Receiver operating characteristic curve (ROC) showing the area under the curve (AUC) for discrimination between MS patients and controls for the concentration levels of pro-and anti-inflammatory factors tested (data with statistical significance are shown). GM-CSF: Granulocyte macrophage-colony stimulating factor; ICAM-1: Intercellular adhesion molecule-1; IFN: Interferon; IL: Interleukin; CXCL: C-X-C Motif Chemokine Ligand; CCL: C-C Motif Chemokine Ligand; TNF-a: Tumor necrosis factor alpha.

**Figure 3 biomedicines-11-00906-f003:**
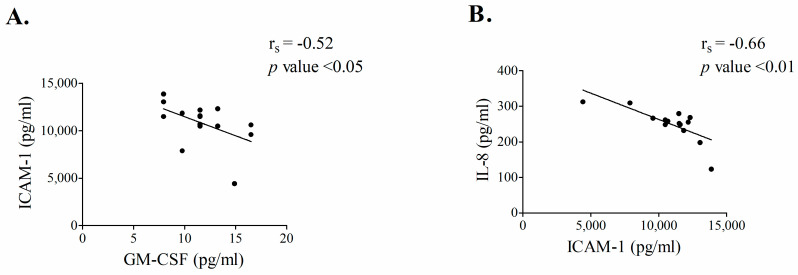
Correlation between the concentration levels of pro-inflammatory mediators upon astrocytic activation followed by IgG stimulation. A negative correlation was revealed between ICAM-1 concentration and GM-CSF (**A**) and between ICAM-1 and IL-8 concentration levels (**B**). Values are expressed as pg/mL. Data with statistical significance are shown.

**Figure 4 biomedicines-11-00906-f004:**
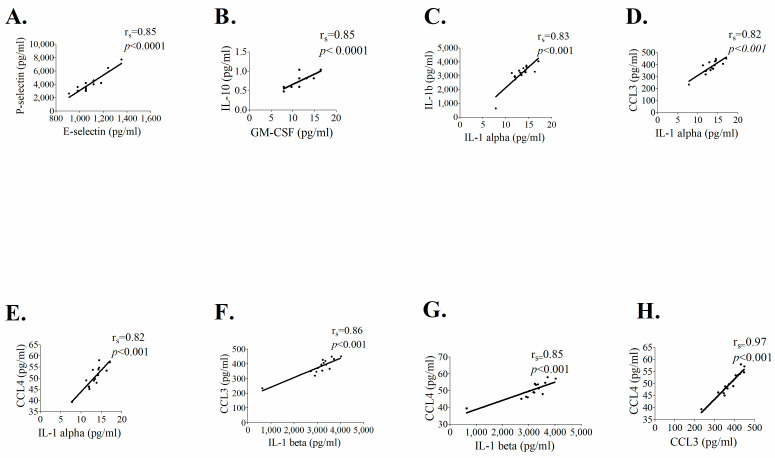
Correlation between the concentration levels of pro-inflammatory mediators upon astrocytic activation followed by IgG stimulation. A strong positive correlation was revealed between E-selectin and P-selectin concentration levels (**A**), between GM-CSF and IL-10 (**B**), IL-1a and IL-1b (**C**), IL-1a and CCL3 (**D**), IL-1a and CCL4 (**E**), IL-1b and CCL3 (**F**), IL-1b and CCL4 (**G**), and between CCL3 and CCL4 concentration levels (**H**). Values are expressed as pg/mL. Data with statistical significance are shown.

**Figure 5 biomedicines-11-00906-f005:**
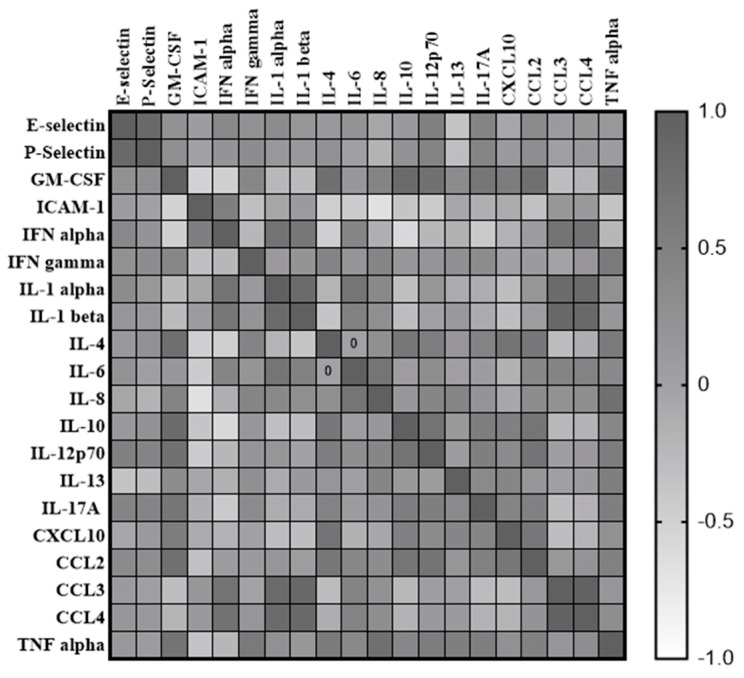
Correlation matrix between the concentration levels of pro- and anti-inflammatory factors upon astrocytic activation followed by IgG stimulation. All the correlation analyses were performed using the Spearman correlation coefficient. The color-coded R values range between −1 (white color) and 1 (gray color), as provided on the heat map.

**Figure 6 biomedicines-11-00906-f006:**
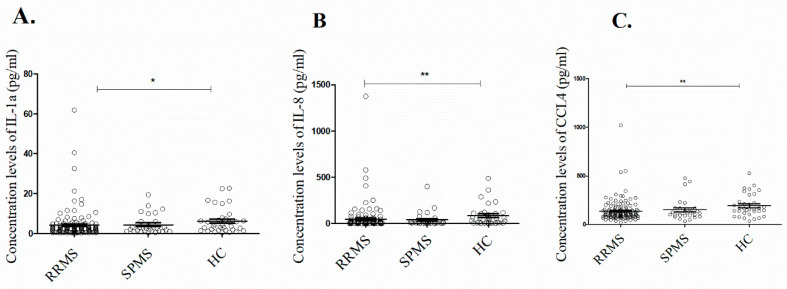
Analysis of the concentration levels of pro-inflammatory molecules IL-1a (**A**), IL-8 (**B**), and CCL4 (**C**) among RRMS, SPMS, and HCs. Bars represent the mean ± SEM. The Kruskal–Wallis test was performed for the analysis, and Dunn’s test for the multiple comparisons (* *p* < 0.05, ** *p* < 0.01) (only data with statistical significance are shown).

**Figure 7 biomedicines-11-00906-f007:**
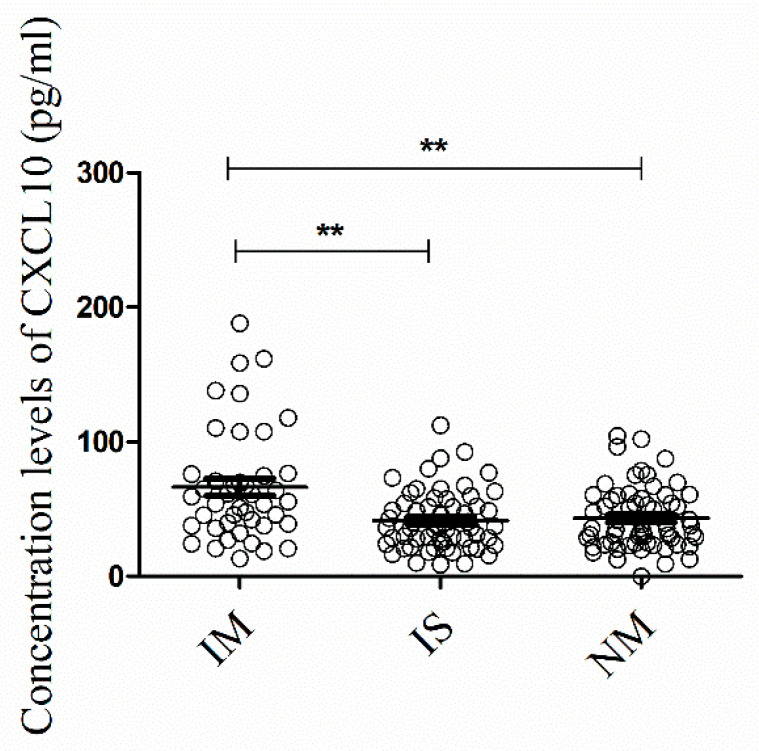
Concentration levels of CXCL10 in regard to medication. MS patients receiving immunomodulatory drugs had elevated levels of CXCL10 compared to patients receiving immunosuppressive drugs or no medication. Bars represent the mean ± SEM. The Kruskal–Wallis test was performed for the analysis (** *p* < 0.01). IM: immunomodulatory drugs; IS: immunosuppressive drugs, NM: no medication.

**Table 1 biomedicines-11-00906-t001:** Demographic, clinical, and laboratory findings of patients positive in antibodies against coagulation factors.

MS Patients	Gender	Age	Disease Duration	Disease Course	EDSS	MSSS	Medication	Laboratory Findings
−1	F	41	23	RRMS	4.0	2.78	Interferon	Anti-FXa
−2	F	59	30	RRMS	2.5	1.19	-	Anti-FVIIa
−3	M	58	24	SPMS	6.0	5.03	-	Anti-plasmin
−4	F	47	21	SPMS	4.0	2.97	-	Anti-plasmin, anti-FXa, anti-FXII
−5	F	65	25	RRMS	3.0	1.56	-	Anti-plasmin, anti-FXa
−6	M	39	12	RRMS	3.0	3.25	Interferon	Anti-plasmin, anti-FXa
−7	F	63	19	PPMS	6.5	6.59	-	Anti-plasmin, anti-FXa, antithrombin
−8	M	33	13	RRMS	3.0	3.05	Fingolimand	Anti-plasmin, anti-PT, anti-FXII, anti-PC
−9	M	62	10	SPMS	7.0	8.92	-	Anti-FXII
−10	F	51	11	RRMS	5.0	5.82	Fingolimand	Anti-PC
−11	M	32	2	RRMS	3.5	7.98	Interferon	Antithrombin
−12	F	53	3	CIS	1.0	1.77	-	Anti-PT
−13	M	40	3	RRMS	3.0	6.81	Fingolimand	Antithrombin
−14	M	36	11	RRMS	3.5	4.21	Fingolimand	Anti-FVIIa
−15	F	28	1	RRMS	3.0	7.93	Interferon	Anti-FXII

MS: Multiple Sclerosis; F: female; M: male; CIS: clinically isolated syndrome; RRMS: Relapsing-remitting multiple sclerosis; SPMS: Secondary progressive multiple sclerosis; PPMS: primary progressive multiple sclerosis; EDSS: expanded disability status scale; MSSS: multiple sclerosis severity scores; PT: prothrombin; PC: protein C.

**Table 2 biomedicines-11-00906-t002:** Expression of pro- and anti-inflammatory molecules in cell lysates from disease and control groups.

	U87 Cells upon Stimulation with MS IgG Fractions		U87 Cells upon Stimulation with HC Samples			
Pro- or Anti-Inflammatory Molecules	Mean (pg/mL)	SEM	Mean (pg/mL)	SEM	*p* Value	AUC (95% CI)
E-selectin	1089	28.12	702.3	15.90	<0.0001	1.00
P-selectin	4108	349.5	7755	343.7	<0.0001	0.95 (0.88–1.0)
GM-CSF	11.69	0.7272	8.435	0.7792	=0.0050	0.76 (0.59–0.94)
ICAM-1	10,790	586.4	9087	616.3	=0.0137	0.77 (0.59–1.95)
IFN-gamma	2.459	0.0835	2.878	0.1255	=0.0059	0.80 (0.62–1.98)
IL-1 alpha	13.30	0.5678	9.574	0.4023	<0.0001	0.93 (0.82–1.0)
IL-1 beta	3120	197.9	2706	115.0	=0.0094	0.78 (0.60–1.96)
IL-4	12.75	0.8069	14.96	1.467	=0.19	-
IL-6	394.5	19.26	447.5	13.91	=0.036	0.73 (0.54–0.92)
IL-8	251.1	11.61	159.6	4.82	<0.0001	0.93 (0.80–1.0)
IL-10	0.7533	0.0480	0.7469	0.0717	=0.94	-
IL-12p70	7.781	0.4429	6.933	0.2768	=0.12	-
IL-13	2.644	0.1391	1.893	0.1619	=0.0016	0.77 (0.58–0.96)
IL-17a	2.787	0.0781	3.446	0.1063	<0.0001	0.91 (0.79–1.0)
CXCL10	2.807	0.1334	2.001	0.2249	=0.0042	0.78 (0.62–0.95)
CCL2	17.08	0.4888	13.13	0.5012	<0.0001	0.89 (0.77–1.0)
CCL3	380.8	14.69	296.8	15.87	=0.0013	0.85 (0.71–0.99)
CCL4	50.30	1.294	39.19	1.708	<0.0001	0.91 (0.81–1.0)
TNF-a	22.37	0.6531	15.77	0.3263	<0.0001	0.98 (0.93–1.0)

Data are expressed as the mean (±SEM) of the concentration for each study molecule. MS: Multiple sclerosis; HC: Healthy control; SEM: Standard error of the mean; IgG: Immunoglobulin G; GM-CSF: Granulocyte macrophage-colony stimulating factor; ICAM-1: Intercellular adhesion molecule-1; IFN: Interferon; IL: Interleukin; CXCL: C-X-C Motif Chemokine Ligand; CCL: C-C Motif Chemokine Ligand; TNF-a: Tumor necrosis factor alpha. The Mann-Whitney test was performed for the analysis.

**Table 3 biomedicines-11-00906-t003:** Demographic and clinical profile of participants.

Features	MS Patients (*n* = 167)	HCs (*n* = 40)	*p* Value
**Gender** Female/Male	110/57	21/19	0.20
**Age in years** Mean ± SD Min–Max	48 ± 13 21–80	45 ± 10 24–63	0.29
**Disease course** (CIS/RRMS/SPMS/PPMS)	1/130/29/7	N/A	
**Disease Duration (years)** Mean ± SD	15.3 ± 8.9	N/A	
**EDSS** Median (interquartile range) Mild: 0–3.0 [*n* (%)] Moderate: 3.5–5.5 [*n* (%)] Severe: 6.0–9.5 [*n* (%)]	3.5 (2.5–5.5) 75 (45.0) 57 (34.0) 35 (21.0)	N/A	
**MSSS** Median (interquartile range) Benign MS: 1–2 [*n* (%)] Severe MS: 7–10 [*n* (%)]	3.9 (2.4–5.5) 20 (12.0) 20 (12.0)	N/A	
**Medication [*n* (%)]** Interferon beta-1a or -1b Natalizumab Fingolimod Other * None	40 (24.0) 18 (11.0) 26 (16.0) 24 (14.0) 59 (35.0)	N/A	

MS: Multiple Sclerosis; HCs: Healthy controls; CIS: Clinically Isolated Syndrome; RRMS: Relapsing-Remitting MS; SPMS: Secondary Progressive MS; PPMS: Primary Progressive MS; SD: Standard Deviation; EDSS: Expanded Disability Status Scale; MSSS: Multiple Sclerosis Severity Score; N/A: Not Applicable; * Other types of treatment: Azathioprine, Dimethyl fumarate, Glatiramer acetate, Methotrexate, Mycophenolate, Rituximab, Teriflunomide.

**Table 4 biomedicines-11-00906-t004:** Concentration levels of pro- and anti-inflammatory mediators in patients and HCs.

	MS Patients		HCs		
Pro- or Anti-Inflammatory Molecules	Mean (pg/mL)	SEM	Mean (pg/mL)	SEM	*p* Value
E-selectin	28,320	912.7	29,120	2452	0.91
P-selectin	937,900	58,580	872,400	126,600	0.42
GM-CSF	7.33	2.40	8.74	2.08	0.0102
ICAM-1	333,500	28,960	271,000	36,170	0.53
IFN-alpha	3.20	0.49	3.25	0.65	0.43
IFN-gamma	23.09	2.40	20.19	4.36	0.25
IL-1alpha	4.85	0.65	6.29	1.06	0.012
IL-1beta	5.79	1.02	5.52	1.33	0.68
IL-4	34.92	2.98	31.46	4.88	0.26
IL-6	26.28	5.54	15.90	2.92	0.64
IL-8	55.59	12.34	84.97	19.21	0.003
IL-10	1.95	0.54	1.22	0.23	0.32
IL-12p70	14.84	2.01	13.30	1.44	0.89
IL-13	10.79	1.37	9.37	1.14	0.62
IL-17a	10.31	1.13	8.26	1.13	0.47
CXCL10	48.68	2.40	47.11	4.44	0.98
CCL2	167.3	11.13	156.2	14.94	0.68
CCL3	37.37	2.54	45.70	5.82	0.12
CCL4	150.9	10.48	193.4	19.87	0.0043
TNF-a	50.27	2.55	48.89	4.86	0.96

Data are expressed as the mean (± SEM) of the concentration for each study molecule. MS: Multiple sclerosis; HC: Healthy control; SEM: Standard error of the mean; GM-CSF: Granulocyte macrophage-colony stimulating factor; ICAM-1: Intercellular adhesion molecule-1; IFN: Interferon; IL: Interleukin; CXCL: C-X-C Motif Chemokine Ligand; CCL: C-C Motif Chemokine Ligand; TNF-a: Tumor necrosis factor alpha. The Mann-Whitney test was performed for the analysis.

**Table 5 biomedicines-11-00906-t005:** Concentration levels of pro-inflammatory factors with respect to disease courses.

Pro-Inflammatory Factors	Mean Index (±SEM)			Kruskal-Wallis Test *p* Value	Dunn’s Multiple Comparisons Test		
	RRMS	SPMS	HCs		RRMS-HCs	SPMS-HCs	RRMS-SPMS
					*p* Value		
GM-CSF	8.82 (±3.17)	3.47 (±2.54)	8.74 (±2.08)	0.03	ns	ns	ns
IL-1a	4.27 (±0.67)	4.48 (±0.89)	6.29 (±1.06)	0.02	*	ns	ns
IL-8	47.85 (±13.01)	40.39 (±15.08)	84.97 (±19.21)	0.007	**	ns	ns
CCL4	139.4 (±10.18)	152.2 (±20.66)	193.4 (±19.85)	0.007	**	ns	ns

SEM: Standard error of the mean; RRMS: Relapsing-Remitting MS; SPMS: Secondary Progressive; HCs: Healthy controls; GM-CSF: Granulocyte macrophage-colony stimulating factor; IL: Interleukin; CCL: C-C Motif Chemokine Ligand; ns: not significant; *p* < 0.05 (*); *p* < 0.01 (**). The Kruskal–Wallis test was performed for the analysis, and Dunn’s test for the multiple comparisons.

## Data Availability

The data that support the findings of this study are available from the corresponding author upon reasonable request.

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
