# Peer review of "A Preclinical Investigation on the Role of IgG Antibodies against Coagulant Components in Multiple Sclerosis"

_biomedicines, 2023, doi:10.3390/biomedicines11030906_

Round 1

Reviewer 1 Report

Thank you very much to allowing me to review the article entitle “A preclinical investigation on the role of IgG antibodies against coagulant components in Multiple Sclerosis.” (biomedicines-2258602).

The aim of this work is to enlighten the role of IgG antibodies against coagulation components by performing a preclinical study, analyzing the astrocytic activation by purified IgG antibodies derived from MS patients, and assessing their possible pro-inflammatory effects using a bead-based multiplexed immunoassay system.

This is a second paper in the same line.

The abstract and the introduction are informative and well structured. But the number of cases and control, could be in the summary, not only the serum samples.

They study 15 MS and 14 healthy participants, why you used this combination?
Normally when the sample size of the cases is small is 1:2, two controls by one case.
Are matched the control by sex and age?

The results are well presented, but the duration of the disease between the cases varies a lot, and Could be the duration of the disease affect the results?
May be the conclusion is too strong in relation with the result obtained.

Author Response

  • The number of cases and controls could be in the summary, not only the serum samples.

The number of cases and controls is now included in the summary as well: “…analyzing the astrocytic activation by purified IgG antibodies derived from 15 MS patients, and assessing their possible pro-inflammatory effects using a bead-based multiplexed immunoassay system. The results were compared with those obtained following astrocyte treatment with samples from 14 age- and gender-matched healthy donors…”

  • You study 15 MS and 14 healthy participants, why you used this combination? Normally when the sample size of the cases is small is 1:2, two controls by one case.

As mentioned in the manuscript, this was a preliminary investigation assessing the potential pro-inflammatory role of IgG antibodies against coagulation components in MS. This was the second phase of a larger study that aimed to analyze the coagulation-inflammation interplay from the perspective of IgG antibodies against clotting factors.

In the complete study, we selected samples from 167 MS patients and 40 age- and gender-matched healthy participants and the results of the first phase (screening for the presence of IgG against FVIIa, thrombin, prothrombin, FXa, FXII, plasmin, and protein C in the two study cohorts) have been published (M. S. Hadjiagapiou et al., 2022: Antibodies to blood coagulation components are implicated in patients with multiple sclerosis).

Because of the limited budget available as well as the extensive techniques involved (especially during the second phase of the preclinical study: IgG purification; endotoxin removal; assessment of endotoxin removal procedure, cell culture, optimization of cell activation using at the beginning coagulant and pro-inflammatory factors and then samples from the patients and controls; lysis of the activated and non-activated cells; expression of pro- and anti-inflammatory factors using the Luminex xMAP technology), we were unable to study the entire group of participants. Thus, we randomly selected to analyze 15 cases and 14 controls.

However, we acknowledge this is a limitation of our study, which we have pointed out in the section on limitations as follows: ‘’…Another limitation was the limited number of samples for the preclinical studies. Since the sample size of MS patients was small, a larger sample size of controls should be analyzed and compared with the results obtained from MS patients to have more valid results or confirm the existing ones…’’

  • Are matched the control by sex and age?

We have assessed the age and gender matching of 15 MS cases and 14 controls and a statement is now included in the first line of the section: 2.1 Study participants as follows: ‘Fifteen MS patients and 14 age- and gender-matched healthy participants (p<0.95 and p=1.0, respectively) were recruited for the preclinical investigation of IgG antibodies against coagulation components.’’

  • About results: the duration of the disease between the cases varies a lot and could be the duration of the disease affect the results?

The duration of the disease along with detection (or not) of IgG antibodies against clotting factors throughout the years constitute the objectives of our upcoming longitudinal analysis. Specifically, we are planning to analyze whether the immunological findings of each patient examined in the study are persistent at different time points of the disease, given the impact of possible alterations of IgG activity on pro-inflammatory expression, and whether duration of the disease could affect the results following a comparison between different periods of the disease.

In the current study, our goal clearly focused on the role of IgG antibodies against different coagulation components and characterizing their effects upon astrocyte activation with purified IgG samples derived from patients.

Our research work however, is an ongoing study of the coagulation-inflammation interplay in MS, analyzing different aspects of the disease and giving new insights that could be valuable for MS research.

In any case, we have feel that the comment addressed is valuable hence we included the suggested comment as part of our limitation in order to give food for thought to our future research: Moreover, since the duration of the disease between the cases studied in our preclinical study varies, a future study is needed to investigate whether the duration of the disease could affect the results following a comparison between different periods of the disease based on IgG activity and expressed pro-inflammatory factors upon astrocytic activation with MS-derived IgG antibodies studied.  

  • Maybe the conclusion is too strong in relation with the results obtained.

We have rewritten the section of the conclusion to avoid any strong statements in the conclusion that may not be related to the results obtained: ‘’Overall, the current study shows for the first time that IgG antibodies against coagulant components play a role in the development of an inflammatory milieu in neuroinflammatory diseases such as MS. Taken together, antibodies derived from MS patients promoted higher levels of concentration for specific pro-inflammatory molecules secreted by astrocytes when compared to samples from healthy participants. Since the study of antibodies warrants further investigation, new findings will shed light on the underlying mechanisms that are activated and cause MS to develop, providing new insights into the disease's prognosis and monitoring, as well as the development of novel therapeutic strategies.’’

Reviewer 2 Report

Reviewer’s Report on Manuscript Entitled:

A preclinical investigation on the role of IgG antibodies against coagulant components in Multiple Sclerosis

The authors investigated the role of IgG antibodies against coagulation components by performing a preclinical study, analyzing the astrocytic activation by purified IgG antibodies derived from MS patients, and assessing their possible pro-inflammatory effects using a bead-based multiplexed immunoassay system. The topic and results are interesting and informative, but the presentation can be further improved. Please see below my comments.

Lines 88-97. The main contributions of the research should be more highlighted here, preferable you may use bullet points to highlight.

In the method section, please add a flowchart showing the workflow of this research.

Lines 132-133., and Lines 144-146. A paragraph should at least have two sentences. Please merge them to the previous paragraphs. Similarly, for lines 181-185, etc. Please check this and adjust accordingly.

Line 193. Some most recent references with applications need to be added which describe the statistical methods in this section:

Line 198. Please define ANOVA. All abbreviations must be defined the first time they are used. Please also add the following paper describing Kruskal–Wallis one-way ANOVA with application:

https://doi.org/10.3390/en14217120

Line 202. Please add the following paper describing ROC and AUC in detail with application in bioengineering: https://doi.org/10.1109/JSEN.2023.3237383

Table 2. Last column. Please also add a figure like in the reference above showing the ROC curves for these AUC values. You figure can have three panels where each panel shows about 5 ROC curved with different colors corresponding to each of the pro- or anti-inflammatory molecules.

Line 203. Please add the following paper describing Spearman correlation coefficient in detail with its application in medicine: https://doi.org/10.3390/app9204284

Lines 256-262. Please re-format the caption.

Figures 1,2,3,4 Have poor resolution. The resolution of the figures must be at least 300dpi. Furthermore, the font size of the texts, numbers, and labels in the figures should be almost the same size as the figure caption.

Line 297. Please use regular font, not bold face.  

Please insert the volume number and page numbers for the references and follow the MDPI guideline for the format and style of references.

Please carefully proofread the manuscript.

Author Response

  • Line 88-97: The main contribution of the research should be more highlighted here, preferable you may use bullet points to highlight.

We have re-written the last paragraph of the introduction, highlighting the contribution of our research project: ‘…Therefore, the following objectives have been established for the current research study:

  • Identification of the role of IgG antibodies against coagulation components in MS. Following a series of in vitroexperiments with purified IgG antibodies, the role of such molecules will be characterized regarding the changes in the expression levels of inflammatory mediators. This will be accomplished by stimulating astrocytes in vitro with purified antibodies from samples of MS patients who showed seropositivity against factor (F)VIIa, thrombin, prothrombin, FXa, FXII, plasmin, and protein C, and comparing the results with those of healthy participants testing negative [17]. First, experimental conditions will be optimized.
  • Analysis of the expression levels of inflammatory and neuroinflammatory mediators in MS patients and comparison with controls.

  • In the method section, please add a flowchart showing the workflow of this research.

A flowchart showing the workflow of our study is presented in the first section of methodology as Figure 1, as suggested. 

  • Lines 132-133 and lines 144-146: A paragraph should have at least two sentences. Please merge them to the previous paragraphs. Similarly, for the lines 181-185, etc. Please check this and adjust accordingly.

The lines have been merged as advised, and the manuscript has been checked and adjusted accordingly.

  • Line 193: Some most recent references with applications need to be added which describe the statistical methods in this section.
  • For the D’ Agostino-Pearson test, the following reference has now been added: doi:10.1111/vcp.12390 describing the sensitivity and specificity of such normality test in a computer-simulation study.
  • For the description of the nonparametric Msnn-Whitney U-test, the following reference has now been added: doi:10.1186/s12936-022-04104-x with application in genetics.
  • The paper https://doi.org/10.1093/bioadv/vbab034 describes the Fisher’s exact test, with application in bioinformatics for the analysis of unknown proteins.
  • For the Kruskal–Wallis (ANOVA) test and ROC/AUC, the references advised in the comment below have been included.

  • Line 198: Please define ANOVA. All abbreviations must be defined the first time they are used. Please also add the follow paper describing Kruskal-Wallis one-way ANOVA with application. doi: 10.3390/en14217120

The ANOVA analysis has been defined: ‘’…Kruskal–Wallis one-way analysis of variance (ANOVA) followed by Dunn's post hoc...’’ and the reference has been added in the text.

  • Line 202: Please add the following paper describing ROC and AUC in detail with application in bioengineering: doi: 10.1109/JSEN.2023.3237383

The suggested paper has been added in Line 202.

  • Table 2: Last column. Please also add a figure like in the reference above showing the ROC curve for these AUC values. Your figure can have three panels where each panel shows 5 ROC curved with different colours corresponding to each of the pro- and anti-inflammatory factors.

A figure has been included in the revised manuscript (Figure 2) similar to the reference suggested, showing the ROC curve for the factors that were significantly different between MS patients and controls.  

  • Line 203: Please add the following paper describing Spearman correlation coefficient in detail with its application in medicine. doi: 10.3390/app9204284

The above reference is now included in the revised manuscript as advised.

  • Line 256-262: Please re-format the caption.

We would like to apologise for omitting the formatting of the caption, due to the large size of Table 2. In order to re-format the caption, we have minimized the Table and we hope this will be acceptable. 

  • Figures 1,2,3,4: Have poor resolution. The resolution of the figures must be at least 300dpi. Furthermore, the font size of the texts, numbers, and labels in the figures should be almost the same size as the figure caption.

All the figures (text, numbers, labels) have been checked and the resolution of each figure is as suggested (using 300dpi).

  • Line 297: Please use regular font, not bold face.

The bold face font in each figure has been changed to regular font (Figure 1-5).

  • Please insert the volume number and the page numbers for the references and follow the MDPI guideline for the format and style references.

The volume number and the page numbers for the references have now been included. Mendeley reference manager has been used, applying the appropriate style for Biomedicines.

Round 2

Reviewer 1 Report

After revisiting the article titled “A preclinical investigation on the role of IgG antibodies against coagulant components in Multiple Sclerosis.” (biomedicines-2258602) and the reviewers' response, I consider that the authors have clarified the doubts about the work carried out.

 Minor comments:

On page 23 the authors say in line 3 “Since the sample size of MS patients was small, a larger sample size of controls” this is not correct, since they use a minimum of one control per case.

On page 24, in the conclusions, the first part conforms to the results obtained, but the second part should be removed as it is speculative, or passed for discussion.

Author Response

Thank you for your comments. We have amended line 3 on pg. 23 and have removed the second part of the conclusions on pg. 24.

Reviewer 2 Report

I would like to thank the authors for addressing my comments. I have a few minor suggestions:

Page 5. Objectives should not be followed by their solutions. Thus, please remove line 4 to line 8. So remove "This will be accomplished by...."

Page 7. Lines 17 and 22. Please do not use the English letter X to stand for multiplication. Use the proper format.

Figures 2 and 4. Please enlarge the font size of the numbers on the x-axis and y-axis values and improve the quality of the figure (minimum 300 dpi resolution).

The conclusion section. Please add one more sentence at the beginning of the conclusion to say what the objective of this work was.

Reference 23. Please correct it as "2021, 1, vbab034" not "2021, 1"

Reference 24. Please correct it as "Eyeblink" not "Eye-Blink"

Reference 25. Please correct it as "2019, 9, 4284" not "2019, 9".

Reference 36. Please correct it as "2022, 9, e000738" not "2022, 9".

Thank you!

Regards,

Author Response

We would like to thank Reviewer 2 for the comments.

All minor comments have been taken into consideration in the revised manuscript attached.